



# The Flexible Modelling Framework for the Met Office Unified Model (Flex-UM, part of the UM 12.1 release)

Penelope Maher[1] and Paul Earnshaw[2]

[1]Department of Mathematics, University of Exeter, Exeter, UK
[2]Met Office, Exeter, UK

**Correspondence:** Penelope Maher (p.maher@exeter.ac.uk)

**Abstract.**

The Met Office Unified Model (UM) is a world-leading atmospheric weather and climate model. In addition to comprehensive simulations of the atmosphere, the UM is capable of running idealised simulations, such as the dry physics Held–Suarez test case, radiative convective equilibrium and simulating planetary atmospheres other than Earth. However, there is a disconnect between the simplicity of the idealised UM model configurations and the full complexity of the UM. This gap inhibits the broad use of climate model hierarchy approaches within the UM. To fill this gap, we have developed the Flexible modelling framework of the UM – Flex-UM – which broadens the climate model hierarchy capabilities within the UM. Flex-UM was designed to replicate the atmospheric physics of the Geophysical Fluid Dynamic Laboratory (GFDL) idealised moist physics aquaplanet model. New parameterisations have been implemented in Flex-UM, including simplified schemes for: convection, large-scale precipitation, radiation, boundary layer and sea surface temperature (SST) boundary conditions. These idealised parameterisations have been implemented in a modular way, so that each scheme is available for use in any model configuration. This has the advantage that we can incrementally add or remove complexity within the model hierarchy. We compare Flex-UM to ERA5 and aquaplanet simulations using the Isca climate modelling framework (based on the GFDL moist physics aquaplanet model) and comprehensive simulations of the UM (using the GA7.0 configuration). We also use two SST boundary conditions to compare the models (fixed SST and a slab ocean). We find the Flex-UM climatologies are similar to both Isca and GA7.0 (though Flex-UM is generally a little cooler, with higher relative humidity, and a less pronounced storm track). Flex-UM has a single InterTropical Convergence Zone (ITCZ) in the slab ocean simulation but a double-ITCZ in the fixed SST simulation. Further work is needed to ensure that the atmospheric energy budget closes to within $1\text{-}2 Wm^{-2}$ , as the current configuration of Flex-UM gains 9-11 $Wm^{-2}$ (the range covers the two SST boundary conditions). Flex-UM greatly extends the modelling hierarchy capabilities of the UM and offers a simplified framework for developing, testing and evaluating parameterisations within the UM.





# 1 Introduction

Earth system models are invaluable tools for predicting future climates and for informing mitigation strategies in response to climate change. However, climate models have grown to be some of the most elaborate computer programs in existence (Jeevanjee et al., 2017). The complexity of Earth system models limit their suitability for investigating the fundamental behaviour of the climate (Polvani et al., 2017) or for making progress in reducing persistent model biases, such as the double-ITCZ or too weak ENSO signal. This is where the power of idealised climate models is realised. Idealised models are used for testing a theory or physical process in which all the redundant complexity (for the problem at hand) has been removed. An idealised climate model seeks to represent the most fundamental components of the climate at the cost of quantitative accuracy and realism (Maher et al., 2019). It is easy to criticise the fidelity of an idealised model, however, their usefulness is not in their accuracy, it is in their simplicity. They offer a simplified test bed for understanding processes, developing new theories, testing hypotheses, bug fixing model code, and developing new parameterisations. They also offer a platform for directly comparing model behaviour, for example the Aqua-Planet Experiment (APE, Williamson et al. (2013)), Tropical Rain belts with an Annual cycle and Continent Model Intercomparison Project (TRAC-MIP, Voigt et al. (2016)) and Radiative Convective Equilibrium Model Intercomparison (RCEMIP, Wing et al. (2018)).

Idealised models offer hope for scientists trying to understand and model the climate by providing an easier to interpret climate system. We then connect idealised models to Earth system models, through a series of intermediate complexity models. This sequence of models, known as a climate model hierarchy, connects our understanding of the Earth system to observations of the real world (Maher et al., 2019).

Having motivated the utility of idealized climate models and how they are used to create a climate model hierarchy, let us consider the current hierarchy capabilities of the UK Met Office atmospheric weather and climate model – the Unified Model (UM). When the UM is coupled to an interactive ocean and sea-ice models (Hewitt et al., 2011), it forms the Hadley Centre Global Environmental Model (HadGEM3-GC3.0, Williams et al. (2018), now in its third version (HadGEM3) and third Global Configuration (GC3.0). This configuration describes the Met Office global climate model (GCM) used to contribute simulations for phase six of the Climate Model Intercomparison Project (CMIP6). When HadGEM3-GC3.0 is coupled to chemistry (UKCA), vegetation (JULES), ocean biogeochemistry (MEDUSA), and land ice (BISICLES) models, it forms the UK's Earth System Model (UKESM1, Sellar et al. (2019)). The UKESM1 is the most sophisticated model within the UK Met Office model hierarchy and was used to contribute Earth system model simulations for CMIP6.

Not all scientific questions or model development tasks require the sophistication of Earth system models, such as UKESM1, or even require an interactive ocean used in the global climate model, such as HadGEM3-GC3.0. For many applications is it useful to assume idealised sea surface boundary conditions. Two examples include replacing the ocean model with prescribed Sea Surface Temperature (SST), following the Atmospheric Model Intercomparions Project (AMIP) (Gates et al., 1999) approach, or by removing the land to create an ocean covered globe without zonal variability in the SST, the so-called "aquaplanet





model". The aquaplanet formulation of the UM was developed by Neale (1999), building on the earlier work of Swinbank et al. (1988), and later published in Neale and Hoskins (2000a, b). This provided the foundation on which the AquaPlanet Experiment (APE) (Blackburn et al., 2013; Williamson et al., 2013) was built. For many modelling centers around the world, the aquaplanet configuration is a benchmark model for testing model performance (Stevens and Bony, 2013; Voigt and Shaw,

2015) and for implementing new parameterisations. Indeed the aquaplanet UM has been illuminating for studying tropical convection, for example the Madden–Julian Oscillation (Woolnough et al., 2001; Inness et al., 2001), the double-ITCZ (Talib et al., 2018), atmospheric tides and the diurnal cycle (Woolnough et al., 2004), and the sensitivity of convectively coupled equatorial waves to entrainment (Peatman et al., 2018).

At the most idealised end of the UM model hierarchy is the "idealised UM". This model configuration broadly describes

a number of idealisations including the Held-Suarez Newtonian relaxation (Held and Suarez, 1994; Mayne et al., 2014b), radiative convective equilibrium (Holloway and Woolnough), and exo-planet configurations (Mayne et al., 2014a; Lines et al., 2018; Boutle et al., 2017; Sergeev et al., 2020). The global configuration of the idealised model has also been broadly used for testing and developing the current dynamical core of the UM (ENDGAME) (Wood et al., 2014).

When viewing all the possible UM configurations through the climate model hierarchy lens – the idea that simple models

are connected to sophisticated models via incremental steps – a large gap emerges between the simplified parameterisations of the idealised UM (e.g. the Held-Suarez model), and the sophisticated parameterisations within the comprehensive UM. Specifically, there is a gap in the intermediate complexity parameterisations of the atmosphere. The purpose of this paper is to narrow this gap in the UM climate model hierarchy using the Flexible framework for the UM (Flex-UM). In doing so, we will highlight the utility of Flex-UM and motivate possible directions for research and model development.

Flex-UM was designed so that it can be directly compared to a benchmark intermediate complexity climate model, the Geophysical Fluid Dynamic Laboratory (GFDL) simplified moist physics aquaplanet (Frierson et al., 2006, 2007). Specifically, Flex-UM can be configured as a slab ocean aquaplanet, with a grey radiation scheme, simplified Betts-Miller convection scheme and simple boundary layer scheme, we describe these in detail in Section 2. Flex-UM is not an independent version of the UM. Flex-UM simply describes a collection of new intermediate complexity parameterisations that have been incorporated into the

UM. Each of these new parameterisations can be switched on or off depending on the use case. This implementation allows for a broad range of intermediate complexity configurations within the UM model hierarchy.

The reanalysis data and model configurations are described in Section 3. In Section 4 we compare the model climatologies for two different SST boundary conditions and then validate these against reanalysis. The first SST boundary condition we consider is the slab-ocean aquaplanet configuration in Section 4.1, where we validate Flex-UM against the similarly configured Isca

climate model. The second SST boundary condition we consider is the fixed-SST aquaplanet configuration, where we compare the behaviour of Flex-UM against the comprehensive UM in Section 4.2. We then evaluate the slab ocean configuration of Flex-UM against the comprehensive UM in Section 4.3. We then summarise the new features of Flex-UM in Section 5 and motivate possible use cases for the new capabilities within the UM model hierarchy.



## 2 Flex-UM Model Description

Flex-UM is a collection of parameterisations designed to fill the gap between the idealised UM and the full complexity of the UM. These parameterisations have been incorporated into the model in a modular way so that the model can be easily changed to build different configurations. Flex-UM is not a static model configuration, rather it is flexible to the design choices of the user. The inspiration for the default configuration of Flex-UM is the GFDL moist aquaplanet model (Frierson et al., 2006), commonly referred to as Frierson's model, which uses simplified physical parameterisations on top of an existing dynamical core. In developing Flex-UM, we started with the Global Atmosphere 7.0 (GA7.0, Walters et al. (2019)) configuration of the UM, which is the atmospheric component of the coupled GC3.0 configuration. The GA7.0 dynamical core ENDGame (Walters et al., 2019) is used in Flex-UM without alteration to the code. The only variations to the default ENDGame configuration for Flex-UM is the model resolution (described in Section 3) and resolution dependent settings (e.g. the model timestep).

In developing Flex-UM, each parameterisation of GA7.0 was either turned off (as it is not needed), re-configured to be more idealised, or replaced with a new idealised parameterisation. These adaptations retain the existing code base and allow for easy transitions between idealised and more complex model configurations. There are five key components of the default Flex-UM configuration: radiation, large-scale precipitation, convection, boundary layer, and SST forcing. In the remainder of this section we will describe each key component and point to the relevant literature for further details.

The default GA7.0 radiation scheme SOCRATES (Manners et al., 2021; Edwards and Slingo, 1996) is turned off in Flex-UM and a new grey radiation scheme has been developed. The grey radiation scheme uses an idealised optical depth to approximate the atmospheric water vapour structure and uses a two-stream approximation to provide an infrared cooling effect through the depth of the atmosphere, described in more detail in Section 2(b) of Frierson et al. (2006). In principal, SOCRATES could be adapted to mimic the grey radiation scheme, and this will be considered for later configurations of Flex-UM to explore the impact of the radiation parameterisation on the model behaviour. The incoming solar radiation for Flex-UM is idealised, so that the radiation directly warms the surface and is a function of latitude only. There is no absorption of the solar radiation in the atmosphere, and there is also no seasonal or diurnal cycle. The details of this forcing are described in section 2(b) of Frierson et al. (2006). Unlike SOCRATES, the grey radiation scheme does not account for the radiative impact of aerosols, trace gases, and ozone. Furthermore, Flex-UM does not include clouds or their radiative impact.

The second key component of Flex-UM is the resolved precipitation (also known as large-scale precipitation). The single-moment microphysics scheme in GA7.0 is turned off and is replaced with a simplified large-scale condensation scheme, described in section 2(e) of Frierson et al. (2006). This scheme generates resolved precipitation when a grid-box is super-saturated. Precipitation falls out of the grid box immediately and is re-evaporated below. The resolved precipitation only reaches the surface if each subsequent layer below is also saturated.

The Gregory-Rowntree convection scheme in GA7.0 is turned off and replaced with the simplified Betts-Miller scheme of Frierson (2007), an idealised version of the Betts-Miller convection scheme (Betts, 1986; Betts and Miller, 1986). The simplified Betts-Miller scheme is an adjustment style convection scheme, where an unstable atmosphere is stabilised via adjustment of the moisture and temperature profiles to reference profiles, see Section 2 of Frierson (2007) for details. The





simplified Betts-Miller scheme that we developed for Flex-UM has already been included in a recent convection scheme comparison by Hwong et al. (2021).

The fourth key component of Flex-UM is the boundary layer scheme. The boundary layer scheme of GA7.0 is adapted to create a simplified Monin-Obukhov boundary layer, see Section 2(d) of Frierson et al. (2006) for details. The existing boundary

layer code was modified, rather than replaced, to make use of the existing vertical diffusion code. The diffusion coefficients from GA7.0 are replaced with those defined in Section 2(d) of Frierson et al. (2006). The bulk Richardson number in Frierson et al. (2006), see their Equ. 15, is replaced with the local Richardson number defined Equ. 2.11 of Smith (1990) for consistency with the existing boundary layer code of the UM.

The final key component of Flex-UM is the SST forcing. Two new SST aquaplanet configurations have been added to the

code base. The first aquaplanet configuration developed for Flex-UM is a zonally-symmetric and time-invariant SST profile from equator to pole. The SST forcing is a second order Legendre polynomial of the form: $T_{SST} = T_0 - \frac{\Delta T}{3}\left(3\sin^2\theta - 1\right)$ where $T_0 = 285$ K and $\Delta T = 40$ K. This SST profile and its default values are the same as in the Frierson model and Isca. The SST is zonally symmetric, only varying in latitude, and is a maximum at the equator. Unlike the commonly used APE profiles, this SST profile is not set to $0°$ C equatorward of $60°$ in each hemisphere, rather the SST profile described above is

bounded between $258°$ K at the poles and $298.33°$ K at the equator. In the remainder of the paper, this SST forcing profile will be referred to as the "fixed-SST" configuration. The fixed-SST profile was used in early development of the Flex-UM parameterisations, as removing SST feedbacks simplified the debugging and validation process.

The second SST configuration is a slab ocean aquaplanet, an idealised single-layer ocean model where atmospheric fluxes interact with SST in the vertical but there is no horizontal ocean transport (i.e. no Q-fluxes). The slab ocean is incorporated into

Flex-UM by adapting the land surface scheme JULES (Best et al., 2011). JULES computes the surface energy exchange over the ocean when a coupled ocean is not in use. In GA7.0, the surface temperature is not allowed to vary from the sensible heat transfer. However, this option has been added to JULES to allow the surface temperature to vary while maintaining consistency with the UM boundary layer scheme. The initial SST profile for the slab ocean is the fixed-SST profile described above. After initialisation, the slab ocean is free to evolve. The implementation of the slab ocean is consistent with the slab ocean described

in Section 2a) of Frierson et al. (2006). The slab ocean we developed for Flex-UM is already in use, for example for terrestrial exoplanets (Boutle et al., 2017).

In summary, the Flex-UM default configuration has been designed following the GFDL moist aquaplanet model. Each of the parameterisations have been implemented in a modular way so that each variant can be used in any suitable UM configuration. The development of Flex-UM enables a broader use of climate model hierarchies for the UM, allowing us to not only better

understand the behaviour of the UM, but also to better understand the climate system. The adaptations and newly implemented parameterisations of Flex-UM are in the process of being scientifically and technically reviewed. The code is intended to be available for the next release of the UM (UM version 12.1, release expected November 2021).



## 3 Model configurations and data

### 3.1 Model set-up

The newly implemented and adapted parameterisations within Flex-UM were designed based on the benchmark Frierson (2007) version of the slab ocean GFDL moist aquaplanet. The Isca modelling framework (Vallis et al., 2018) is an outgrowth

of the GFDL aquaplanet, sharing the spectral core and many of its parameterisations. In Section 4.1 we compare the slab-ocean configurations of Flex-UM and Isca. In order to directly compare these two models, Flex-UM was configured to be as similar as possible to the Isca default configurations: an albedo of 0.31, no diurnal or seasonal cycle, constant incoming solar ration of 1360 $Wm^{-2}$ and a 2.5 m slab ocean depth. One difference is that Isca has solar absorption in the grey radiation scheme (consistent with Frierson (2007) while Flex-UM does not (consistent with Frierson et al. (2006). This is discussed in more

detail in Section 4.1.

Flex-UM is then compared to two aquaplanet variants of the UM using the standard GA7.0 atmospheric physics. The first is the fixed-SST configuration (shown in Section 4.2) and the second is the slab ocean configuration (shown in Section 4.3). The purpose of these comparisons is to validate Flex-UM and to evaluate the behaviour of the UM in both the reduced complexity and comprehensive configurations. The key difference between the Flex-UM and GA7.0 simulations are the sophistication of

the atmospheric parameterisations used. Compared to Flex-UM, the GA7.0 parameterisations are more comprehensive. The parameterisations and the default values used for the GA7.0 simulations are described in Walters et al. (2019). In particular GA7.0 uses: the Gregory–Rowntree mass flux convection, the SOCRATES radiation scheme, a comprehensive boundary layer, the PC2 cloud scheme, and has non-orographic gravity waves. The Flex-UM and GA7.0 simulations use the same dynamical core, but have very different atmospheric physics. As such, their simulations are not expected to give the same model climatolo-

gies. These differences should not be viewed as undesirable, rather these are deliberate simplifications to the parameterisations and the key motivation for developing Flex-UM, that is, to build a reduced complexity version of the UM.

### 3.2 Data

The Flex-UM model climatologies are compared to monthly ERA5 reanalysis climatologies (Hersbach et al., 2020; Copernicus Climate Change Service, 2017), the fifth generation of atmospheric reanalysis from ECMWF. The climatologies are computed

over the January 1979 to December 2020 time period. The horizontal resolution of the data is $0.25° \times 0.25°$ and all available vertical levels are used between 1000-100 hPa (27 vertical levels). The ERA5 surface pressure was used to create a mask for discarding data that had been interpolated to the 1000 hPa pressure level if the surface pressure is less than 1000 hPa. Interpolation is prone is errors in the data and this is especially important in computing the mass stream function.

Two Flex-UM simulations are evaluated in this paper, the fixed-SST and slab-ocean aquaplanet configurations. The Flex-

UM simulations have a horizontal resolution of N48 ($2.8° \times 2.8°$), with 38 vertical levels and a 40 km model top. The model output is post-processed from 38 hybrid levels onto 17 pressure levels between 1000 hPa and 10 hPa. The slab-ocean variant of Flex-UM is compared to a slab-ocean Isca configuration. The Isca simulation has a horizontal resolution of T42 ($2.8° \times 2.8°$), 25 vertical sigma levels which are post processed on 14 pressure levels from the surface up to the model top (approximately





5 hPa). The Flex-UM and Isca simulations were performed with horizontal and vertical resolutions as similar as possible. The GA7.0 simulations have a horizontal resolution of N96 ($1.25° \times 1.875°$), with 85 vertical levels and a 85 km model top. The GA7.0 output is post-processed from 85 hybrid levels onto 17 pressure levels between 1000 hPa and 10 hPa.

The horizontal and vertical resolutions are different between the reanalysis and models, and between the UM simulations
as well. This was by design, as we ran the GA7.0 and Isca experiments at their standard resolutions. Flex-UM was run at a similar resolution to Isca so that the two simulations are as similar as possible. This resolution choice was because the primary goal of Flex-UM is to implement the intermediate complexity parameterisations from Isca into the UM, and then compare the two models. We note that when comparing Flex-UM and GA7.0, the key differences in the climatologies are highly unlikely to be the result of resolution differences, as the Flex-UM model physics is very different to the GA7.0 case. All climatologies
are plotted on their native resolution and data used to make difference plots have been linearly interpolated in the vertical to 12 common levels sets between 1000 hPa and 100 hPa, and horizontally regridded to a common resolution of $2.8° \times 2.8°$(the native resolution of Flex-UM).

The initial conditions for the Flex-UM and Isca simulations are an isothermal atmosphere with near-zero moisture. As such, their simulations require a suitable spin-up. A 10-year spin-up was discarded from the simulations, which is standard for these
types of models, and the models were run for a further 10 years. The GA7.0 simulations do not need a long spin up as the simulations start from initial conditions using a climate simulation that was spun-up for 30 years (as standard for GA7.0). However, for consistency with the other simulations, a 20 year run was performed and the first 10 years discarded as spin up. We note that the ERA5 dataset used in this study is 42 years long and the model runs are 20 years. In general it is good practice to use a common time period between datasets, however, as the purpose of this paper is to validate an idealised model, the
simplifications in the model physics will dominate over any differences in the reanalysis time period. For this reason, we felt there was no need to restrict the ERA5 data to the common 20 year period.

The reanalysis and model simulations will be compared in Section 4. For each dataset, the climatological precipitation and zonal-means of temperature (T), relative humidity (RH), zonal wind (u), and the mass stream function ($\Psi$) will be shown. The RH is computed with respect to water above $0°$ C for all datasets, ice below -23$°$ C for ERA5, ice below -20$°$ C for Isca and
ice below $0°$ C for the UM, and interpolated for temperatures in between for ERA5 and Isca. In this study, we also show the atmospheric energy budget, which is a simple global balance between the radiation lost to space, heat absorbed by the surface, and the heat released at the surface, given by:

$$Net = -LWC + SWA + SH + LH \tag{1}$$

where $Net$ describes the residual from the longwave cooling $LWC = LWUT + LWDS - LWUS$, shortwave absorption
$SWA = (SWDT - SWUT) - (SWDS - SWUS)$, sensible heat (SH) and latent heat (LH) (DeAngelis et al., 2015). The radiative fluxes in the LWC and SWA equations are described in terms of the longwave (LW) and shortwave (SW) fluxes at the surface (S) and top-of-atmosphere (T), and for the upward (U) and downward (D) flux directions. Care needs to be taken in how the $LH$ is computed. It is common to use $LH = L_c P$, where $L_c$ is the latent heat of condensation and $P$ is precipitation ($P = P_{rain} + P_{snow}$). However, it is more accurate to use $LH = L_c P_{rain} + L_s P_{snow}$ or $LH = L_c P_{precip} + L_f P_{snow}$ where





$L_s$ and $L_f$ are the latent heat of sublimation and fusion, respectively. More discussion on this can be found in Pendergrass and Hartmann (2013). For ERA5, $LH$ is computed by treating precipitation and snow separately. The LH in all model simulations is computed using precipitation only ($LH = L_c P$), as the Flex-UM and Isca models do not include snow.

## 4 Evaluating the Flex-UM Climatology

In this section we compare Flex-UM climatologies with ERA5 reanalysis, the simplified climate modelling framework Isca, and two comprehensive UM simulations using the GA7.0 configuration. Two SST aquaplanet boundary conditions are used in this study: a slab ocean aquaplanet for comparing Flex-UM to Isca in Section 4.1 and GA7.0 in Section 4.3, and a fixed-SST aquaplanet for comparing to GA7.0 in Section 4.2. The fixed-SST aquaplanet is constant throughout the simulation and does not interact with the atmosphere. The slab-ocean aquaplanet is free to evolve and exchanges fluxes with the surface in the vertical (there is no horizontal transport).

Before considering the model data, we first present the ERA5 reanalysis climatologies in Fig. 1. The general structure of the zonal-mean climatologies for temperature, relative humidity, zonal wind, mass stream function and global precipitation are relatively well known. The warm and moist tropical tropospheric air ascends near the equator and descends in the drier subtropics within the Hadley circulation. The subtropical jet in the upper troposphere and the barotropic eddy driven jet throughout the troposphere can be seen in the zonal-mean zonal wind plots. Poleward of the subtropics, are the Ferrel and Polar cells in each hemisphere. The global precipitation distribution is dominated by tropical rainfall within the Intertropical convergence zone (ITCZ) near the equator and the South Pacific convergence zone (SPCZ) in the equatorial SH Pacific Ocean region.

     Perhaps less well known is the atmospheric energy budget shown in Fig. 1(e), the area weighted global means are shown in the legend. The radiation lost to space (LWC in blue) peaks in the SH and the radiation gains from solar absorption (SWA in orange) peak in the NH near the equator. Both the LWC and SWA reduce to zero at the poles. The latent heat generated from precipitation (LH in red) shows two peaks for the ITCZ and SPCZ. The sensible heat released from the surface (SH in green) is small at all latitudes. The residual atmospheric energy budget (i.e. the net in Equ. 1) is small but non-zero (-3.6 $Wm^2$) and the shape of the residual zonal-mean mirrors the LH from precipitation.

     In the following sections we directly compare Flex-UM with both Isca and GA7.0, and will make more general comparisons

to ERA5. All of the model simulations in this study are aquaplanets, that do not attempt to model the full complexity of the atmosphere, and so direct differences with ERA5 will not be presented. Rather we will make general comparisons to ERA5 to highlight the realism of the model simulations.

### 4.1 Slab ocean aquaplanet: Flex-UM vs Isca

The motivation for developing Flex-UM was to replicate the idealised parameterisations within the GFDL moist aquaplanet

model. As such, we directly compare Flex-UM with the Isca climate model, which is based off the GFDL moist aquaplanet. The temperature and RH from the slab ocean aquaplanet simulations for Flex-UM and Isca are shown in Fig. 2. Both Flex-UM and Isca have similar temperature structures to ERA5 and to each other. Flex-UM is generally cooler than Isca, especially



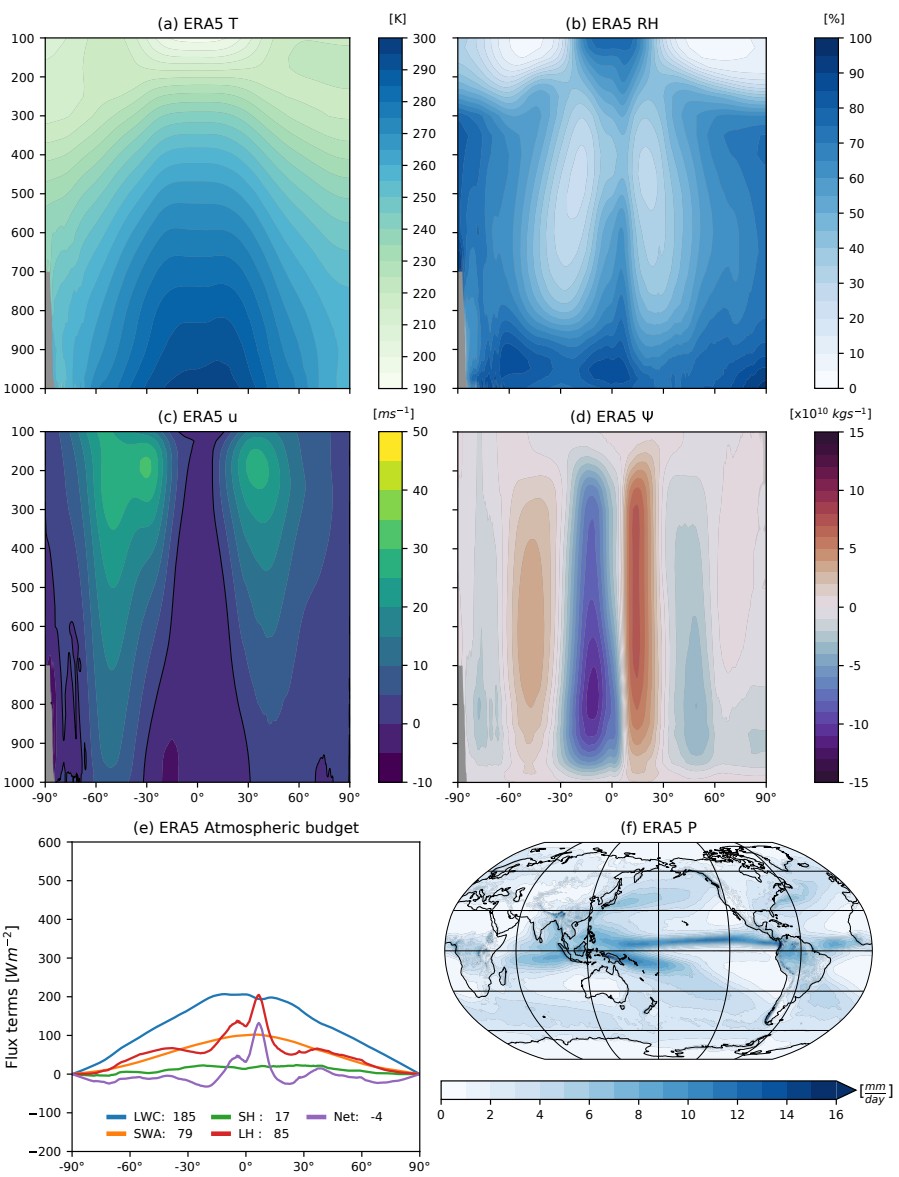

**Figure 1.** ERA5 reanalysis climatologies for: (a) zonal-mean temperature ($T$ in $K$), (b) zonal-mean relative humidity ($RH$ as a %), (c) zonal-mean zonal wind ($u$ in $ms^{-1}$), (d) meridional mass stream function ($\Psi$ in $\times 10^{10} kgs^{-1}$), (e) atmospheric energy budget (see Equ. 1 in $Wm^{-2}$), and (f) precipitation ($P$ in $mmday^{-1}$). Grey contouring in the SH in subplots (a-d) is data masked where the surface pressure is less than the plotted contour level. The black contour in subplot (c) is the zero wind contour. The legend plotted in subplot (e) are the area weighted global averages.

in the upper troposphere and lower stratosphere. The general structure of the RH distribution is broadly similar in Flex-UM and Isca with a few notable differences. Compared to Isca, Flex-UM has a higher RH in the middle and upper troposphere





from the subtropics poleward, and higher RH in the tropical upper tropopause. Compared to ERA5, both Flex-UM and Isca have higher RH at all latitudes. While the difference in RH between Flex-UM and Isca is large in the polar regions, the specific humidity difference is very small in these regions, where their zonal-mean difference is less than 0.5 $gkg^{-1}$ (not shown), owing to the cooler polar temperatures where the specific humidity is small. So while the Flex-UM polar regions might be closer to

5 saturation than the Isca polar regions, the moisture content is too small to make a meaningful difference in the two simulations.

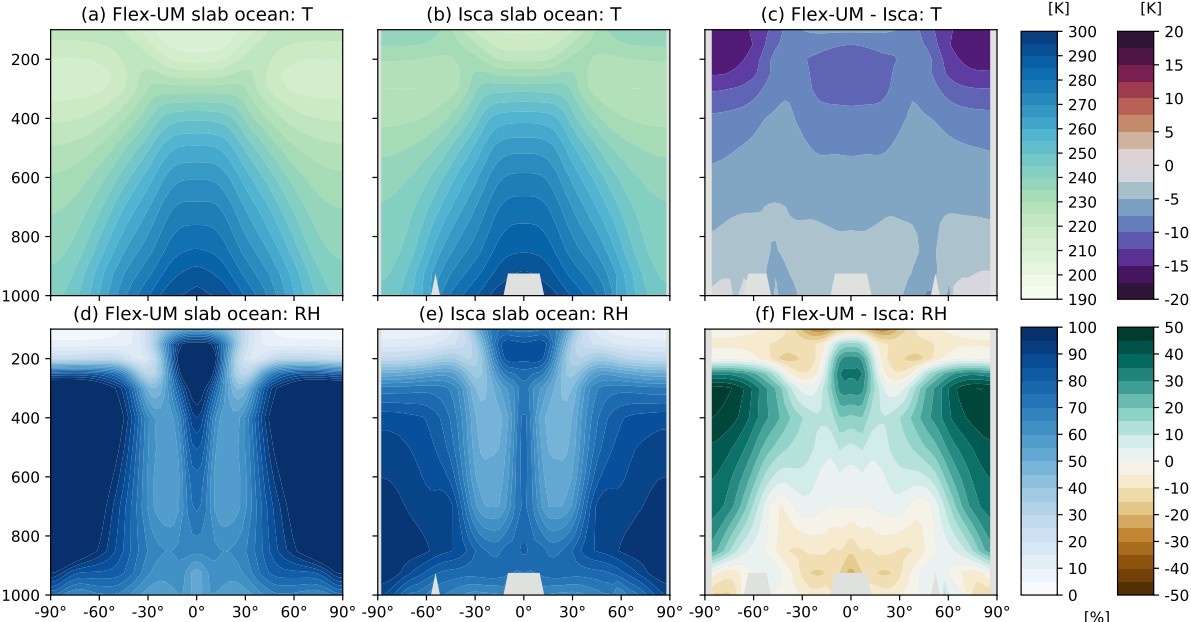

**Figure 2.** Slab ocean aquaplanet model climatologies for Flex-UM (left) and Isca (middle) for temperature ($T$ in $K$, top panel) and relative humidity ($RH$ as a %, bottom panel) on their native model grids. The differences between the models (Flex-UM - Isca) is plotted in the right panels and are interpolated onto the Flex-UM grid. Grey masking near the surface in the middle and right plots occur where the Isca surface pressure is less than the interpolated model level.

The zonal-mean zonal wind and mass stream function from the slab ocean aquaplanet simulations for Flex-UM and Isca are shown in Fig. 3. Flex-UM and Isca both capture the general structure of the zonal wind seen in ERA5 (the subtropical and eddy driven jets are similar). The magnitude of the zonal wind is similar in Flex-UM and Isca, with small differences in the shape of their distributions above 200 hPa. The Flex-UM and Isca mass stream functions are also very similar in shape and cell locations, with Flex-UM a little stronger in both Hadley cells. We note two unrealistic features seen in both Flex-UM

and Isca: the contours of the upper level Hadley cells are not rounded in the upper troposphere as expected, and there is a near surface polar circulation cell seen in the NH in both models that differ in sign. The mass stream function is computed by vertically integrating the zonal-mean meridional velocity ($v$) from the surface to the model top and the horizontal integration of the zonal-mean vertical pressure velocity ($\omega$) from the SH to NH, see equations 6.10-6.11 of Hartmann (1994). The average

of the two integrations is used for $\psi$. The irregular shape of the Hadley cell tops is found in the vertical integration of $v$, which we attribute to the limited number of model levels (38 for Flex-UM and 25 levels for Isca, both interpolated to a common 12





vertical levels between 1000-100 hPa) and horizontal resolution ($2.8°$ lat). The NH near surface polar $\psi$ contours originate from the horizontal integration of $\omega$ and is also likely the result of the course model resolution. As these are integration artifacts, we pay little attention to the differences in these regions.

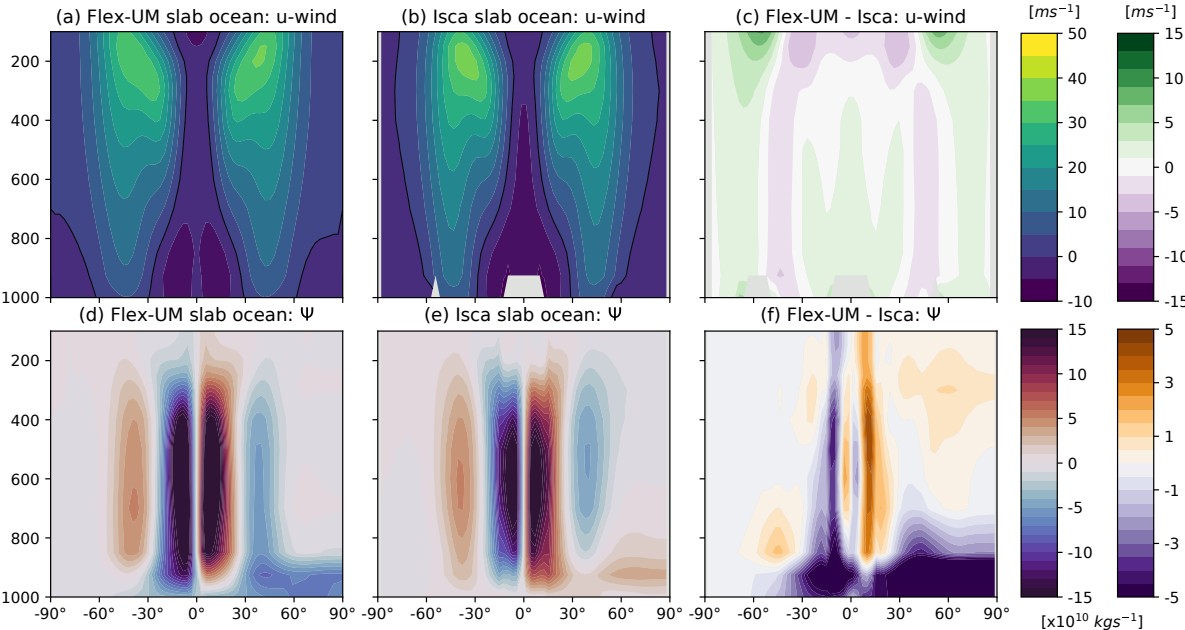

**Figure 3.** As in Fig. 2 but for zonal-mean zonal wind ($u$ in $ms^{-1}$, top panel) and mass stream function ($\Psi$ in $\times 10^{10} kgs^{-1}$, bottom panel).

The precipitation and atmospheric energy budget from the slab ocean aquaplanet simulations for Flex-UM and Isca are
shown in Fig. 4. The precipitation distribution of zonally symmetric aquaplanets with no seasonal cycle are not expected to show the spatial distribution seen in ERA5. Rather, it is more common for the precipitation to peak on the equator (a single ITCZ), or for two peaks to occur on either side of the equator (double ITCZ). Isca has a single ITCZ and a second peak in precipitation in the storm tracks near $40°$ in each hemisphere (seen in Fig. 4(b) and in the LH of Fig. 4(e). The peak equatorial precipitation for Isca is 18 $mmday^{-1}$ and in the storm tracks 5 $mmday^{-1}$. Compared to Isca, Flex-UM has a broader ITCZ
with less equatorial precipitation which peaks at 14 $mmday^{-1}$ on the equator. The storm tracks are less pronounced in Flex-UM, where the subtropical precipitation reduces to 5 $mmday^{-1}$ but does not have a local subtropical minimum or a local storm track maximum.

Compared to ERA5, the precipitation is higher in both Flex-UM and Isca, hence the larger LH component of the atmospheric energy budget Fig. 4(d-e). The shape of the zonal-mean LWC is similar in ERA5, Flex-UM and Isca. The Isca LWC is much
larger than ERA5 and Flex-UM, which is due to more outgoing LW radiation (OLR) and a smaller net surface flux (not shown). The LWC in Flex-UM is similar to ERA5 but a little lower, and each LW radiative flux of Flex-UM is smaller than ERA5 and Isca (not shown). The SH flux is small in ERA5, Flex-UM and Isca. The SWA for Flex-UM is quite different to Isca and ERA5. The globally averaged incoming solar radiation is the same for ERA5, Flex-UM and Isca, while the outgoing SW

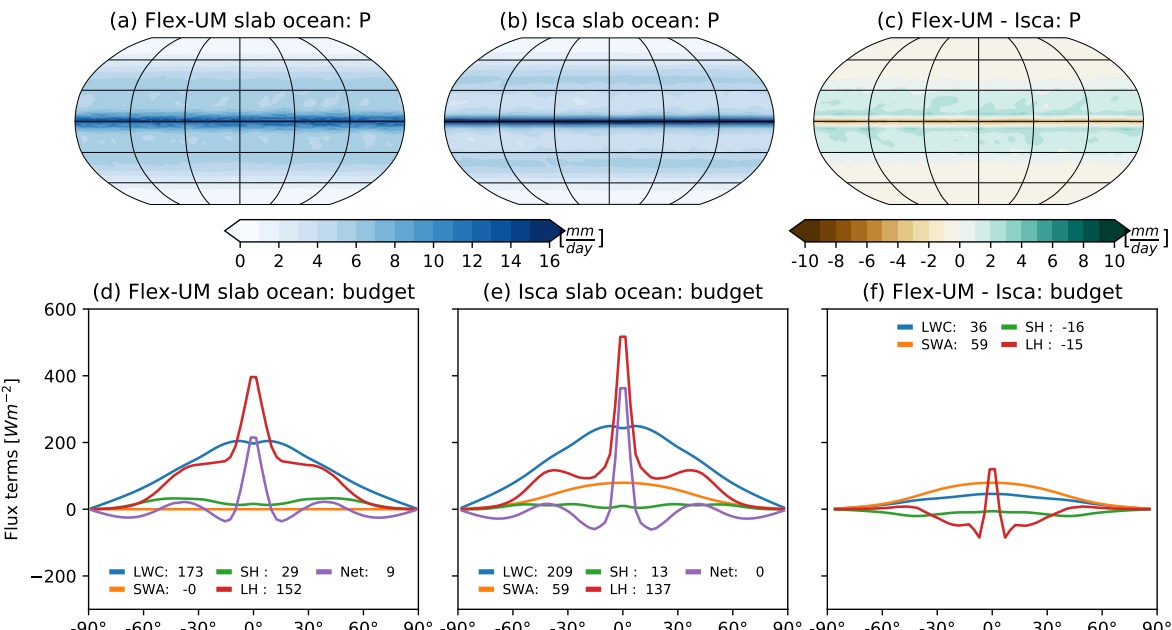

**Figure 4.** As in Fig. 2 but for precipitation ($P$ in $mmday-1$, top panel) and the atmospheric energy budget ($Wm^{-2}$, bottom panel).

flux and surface SW fluxes are all larger for the two models (not shown). For both Flex-UM and Isca, the SW upward flux at the surface and TOA are the same as there is no upwelling solar absorption (not shown). The SWA of Flex-UM is zero as there is no solar absorption included in the model. The net atmospheric budget for Isca is 0.4 $Wm^{-2}$, while for Flex-UM it is 8.8 $Wm^{-2}$. Closing the atmospheric budget will be a priority for later configurations of Flex-UM. The treatment of the

solar absorption within the grey radiation scheme is a key difference between Flex-UM and Isca. In Isca, the downward solar radiation flux depends on the top of atmosphere insolation and optical depth (see Equ. 18 of Frierson (2007)), and the upward solar radiation is a function of the surface albedo and the surface downwelling solar radiation. For Flex-UM, the optical depth is treated as totally transparent so that no solar absorption occurs within the downwelling or upwelling, which is why the SWA is zero in Fig. 4d). This implementation is consistent with the earlier development of the GFDL aquaplanet in Frierson et al.

(2006) which later included shortwave absorption in the downwelling in Frierson (2007). Including downward solar absorption in Flex-UM will be considered for later configurations of Flex-UM.

## 4.2    Fixed-SST aquaplanet: Flex-UM vs GA7

Having validated the slab ocean configuration of Flex-UM by comparison to ERA5 and Isca in Section 4.1, we now compare Flex-UM and GA7.0 for the fixed-SST configuration in this section and the slab ocean configuration in the next section. The

temperature and RH from the fixed-SST aquaplanet simulations for Flex-UM and GA7.0 are shown in Fig. 5. The shape of the Flex-UM and GA7.0 temperature and RH distributions are broadly similar. Flex-UM is cooler throughout most of the





troposphere, especially the tropics between 600-200 hPa, and warmer above 200 hPa. Flex-UM has a higher RH throughout most of the troposphere (800-300 hPa), especially poleward of 50° and within the off-equatorial tropics, and drier near the surface and above 300 hPa. The upper tropospheric tropical peak RH in Flex-UM occurs at a lower altitude compared to Flex-UM and peaks on either side of the equator, which is consistent with a double ITCZ structure (seen more clearly in the

precipitation plots in Fig. 7).

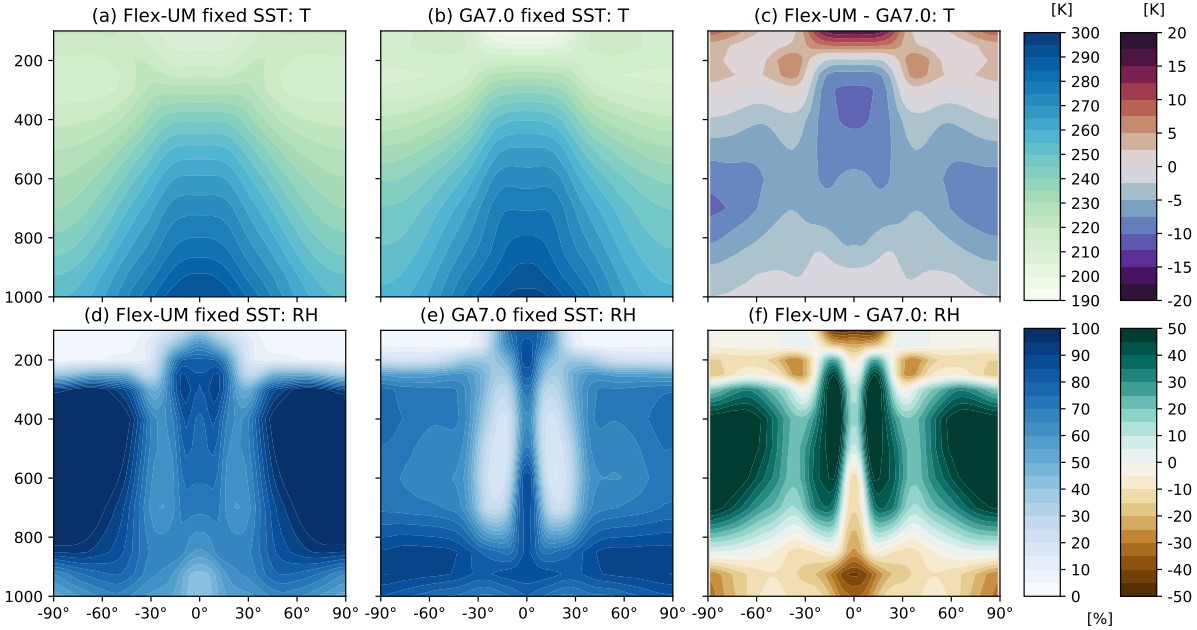

**Figure 5.** Fixed-SST aquaplanet model climatologies for Flex-UM (left) and GA7.0 (middle) for temperature ($T$ in $K$, top panel) and relative humidity ($RH$ as a %, bottom panel) on their native model grids. The differences between the models (Flex-UM - GA7.0) is plotted in the right panels and are interpolated onto the Flex-UM grid.

The zonal-mean zonal wind and mass stream function from the fixed-SST aquaplanet simulations for Flex-UM and GA7.0 are shown in Fig. 6. The Flex-UM zonal wind is located further off the equator than GA7.0, consistent with a double ITCZ structure. The Flex-UM zonal wind is weaker in the tropics and subtropics (45°S-45°N), especially in subtropical jet core. The Flex-UM subtropical jet is more distinct from the eddy-driven jet compared to GA7.0. We also note that Flex-UM and GA7.0

are weakly superrotating aloft, where the equatorial wind is westerly instead of easterly. Superrotation in idealised models is not uncommon, and has previously been found in aquaplanets, shallow-water and Held-Suarez models (Mori et al., 2013; Blackburn et al., 2013; Showman and Polvani, 2010; Lutsko, 2018).

The Hadley circulation in Flex-UM is narrower, shallower, weaker and pushed off the equator in both hemispheres, while GA7.0 is more like ERA5. As seen in the Flex-UM and Isca slab ocean mass stream function plots in Fig. 3d-e), the Flex-UM

fixed-SST simulation has an unrealistic upper Hadley cell shape and the mass stream function in the NH polar region does not



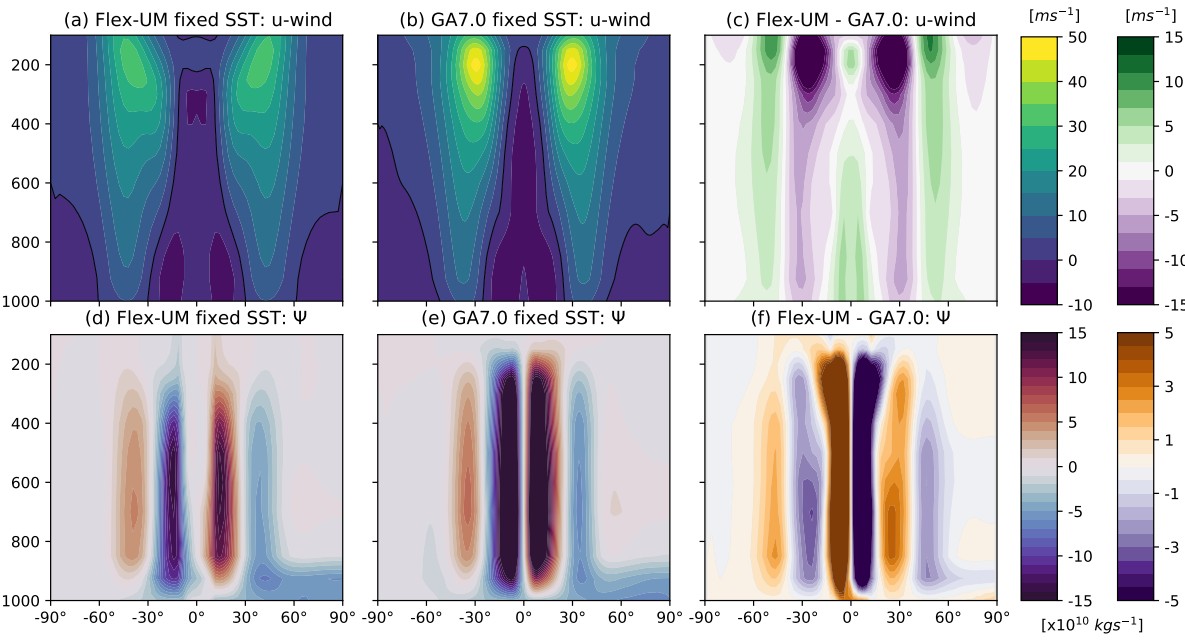

**Figure 6.** As in Fig. 5 but for zonal-mean zonal wind ($u$ in $ms^{-1}$, top panel) and mass stream function ($\Psi$ in $\times 10^{10} kgs^{-1}$, bottom panel).

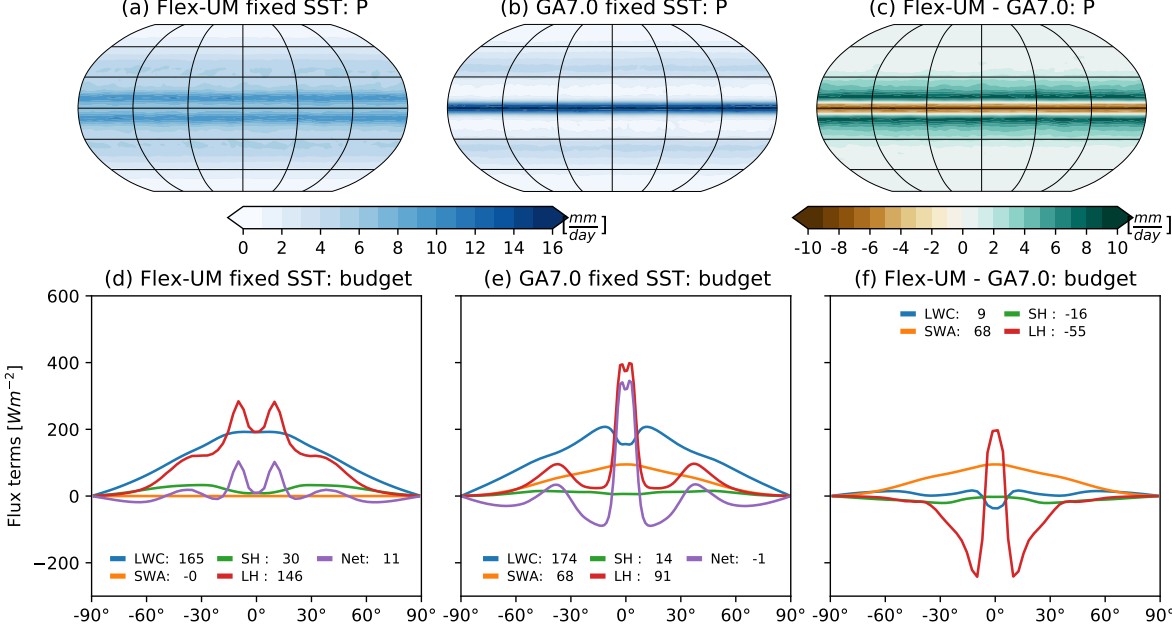

**Figure 7.** As in Fig. 5 but for precipitation ($P$ in $mmday-1$, top panel) and the atmospheric energy budget ($Wm^{-2}$, bottom panel).





represent the polar cell. In GA7.0, the Hadley cell shape is more realistic, which we attribute to higher horizontal and vertical resolution, however the NH near surface $\Psi$ remains unrealistic.

The precipitation and atmospheric energy budget from the fixed-SST aquaplanet simulations for Flex-UM and GA7.0 are shown in Fig. 7. The double ITCZ in Flex-UM is evident in the precipitation map Fig. 7a) and in the zonal-mean LH Fig. 7d).

There is also a small double peak of precipitation in GA7.0, though not separated enough to be considered a double ITCZ. The Flex-UM tropical precipitation peaks are shallower and broader than GA7.0, the storm tracks precipitation is higher, and the subtropical precipitation is higher than expected compared to the storm tracks. Compared to GA7.0, the LH in Flex-UM is considerably larger in the global mean, due to more rainfall (despite the shallower precipitation peaks at the equator). The SWA in Flex-UM is zero, discussed in more detail in Section 4.1, and in GA7.0 the SWA is similar to ERA5 although a little

weaker. The LWC in Flex-UM and GA7.0 are also similar except within the ITCZ where the LWC is reduced, due to reduced OLR due to clouds in GA7.0 which are not modelled in Flex-UM, not shown. The GA7.0 budget closes to within -1.4 $Wm^{-2}$ while the Flex-UM budget does not, with a residual of $10.9Wm^{-2}$.

The double ITCZ seen in the fixed-SST simulations for Flex-UM (and also very weakly in GA7.0) is a common structure seen in aquaplanets (Williamson et al., 2013; Rios-Berrios et al., 2020). The double ITCZ problem is also a well know model

bias in comprehensive GCM where precipitation over the Pacific Ocean is too zonal, see for example Tian (2015); Zhang et al. (2015). The double ITCZ in aquaplanets is known to be sensitive to a number of model choices and is sensitive to feedbacks between the convection, cloud radiative effects and the large-scale circulation. Whether a single or double ITCZ occurs in an aquaplanet model has been shown to depend on the choice of convection scheme and the convection scheme parameters such as entrainment (Möbis and Stevens, 2012). However, the double ITCZ also occurs in comprehensive GCM simulations without

convection schemes where only resolved precipitation occurs (Maher et al., 2018). The appearance of a single or double ITCZ also depends on cloud radiative effects (Harrop and Hartmann, 2016; Popp and Silvers, 2017), and the energy balance near the equator (Kang et al., 2008; Bischoff and Schneider, 2016). It has also been shown the single and double ITCZ can result from no changes in the parameterisaitons but from changes in the model resolution and dynamics core (Landu et al., 2014). Understanding the mechanisms that control the ITCZ and the occurrence of the double ITCZ are long standing problem and

areas of active research.

### 4.3   Comparison of slab: Flex-UM and GA7

Finally, we compare the slab ocean aquaplanet simulations of Flex-UM and GA7.0 and compare the fixed-STT and slab ocean simulations for both models. The temperature and relative humidity for the slab ocean simulations are shown in Fig. 8. The general temperate and humidity structure of Flex-UM in the fixed-SST and slab ocean simulations are similar (compare Fig.

5 and Fig. 8). The primary difference between the Flex-UM simulations is the slab-ocean simulation has a realistic tropical upper troposphere relative humidity (which has a single maximum at a higher altitude in the slab ocean case compared to the fixed-SST case). For GA7.0, the fixed-SST and slab ocean simulations are also very similar for both temperature and relative humidity. The only minor differences are a small weakening in the equatorial relative humidity in the slab ocean case and a small expansion of the subtropical dry regions. The model differences in the slab ocean simulations, Fig. 8(c) and (d), are





similar in shape and magnitude to the fixed-SST model differences, Fig. 5(c) and (d), where Flex-UM is cooler and moister throughout most of the troposphere and drier in the lower levels near the surface.

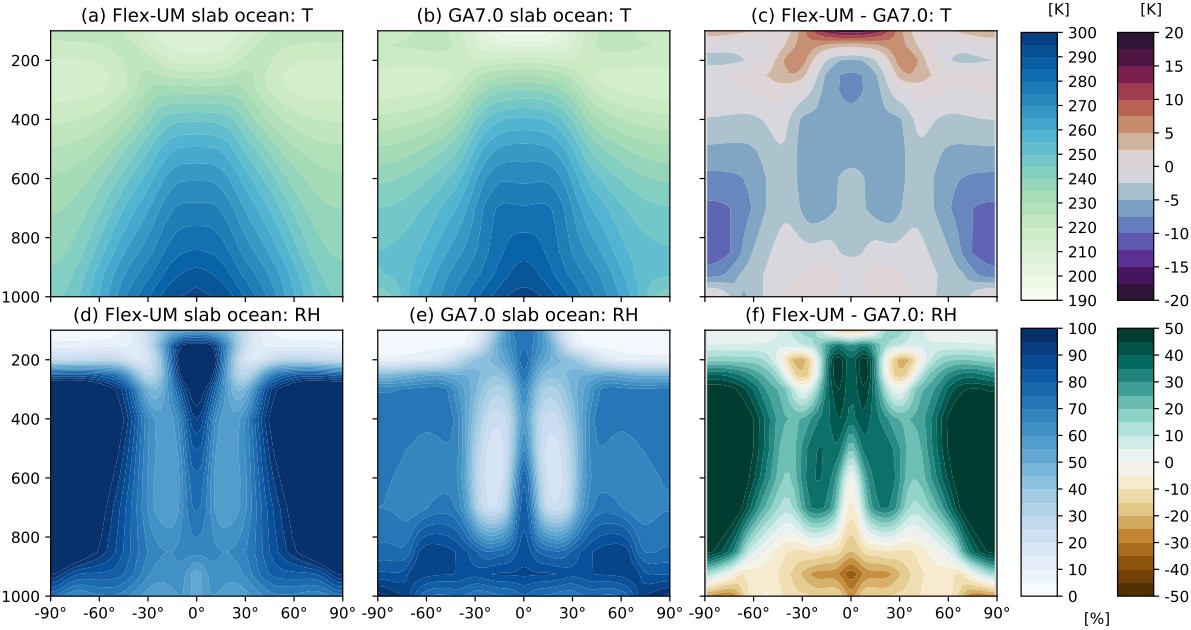

**Figure 8.** Slab ocean aquaplanet model climatologies for Flex-UM (left) and GA7.0 (middle) for temperature ($T$ in $K$, top panel) and relative humidity ($RH$ as a %, bottom panel) on their native model grids. The differences between the models (Flex-UM - GA7.0) is plotted in the right panels and are interpolated onto the Flex-UM grid. Subplots (a) and (d) are identical to Fig. 2 (a) and (d) but are shown here for ease of comparison.

The zonal-mean zonal wind and mass stream function from the slab ocean aquaplanet simulations for Flex-UM and GA7.0 are shown in Fig. 9. The circulation in Flex-UM is more realistic in the slab-ocean case compared to the fixed-SST case, where the jet streams are more distinct, the zonal wind structure is broader, and the Hadley cells are stronger and closer together. The slab ocean and fixed-SST simulations for GA7.0 are very similar. The zonal wind for the Flex-UM and GA7.0 slab ocean simulations differ in the equatorial upper troposphere where GA7.0 has westerlies and Flex-UM has easterlies aloft (i.e. GA7.0 has superrotation while Flex-UM does not, see Fig. 9c)). Like the fixed-SST simulations, the GA7.0 slab ocean simulation has stronger zonal winds and does not extend as far poleward compared to Flex-UM. The circulation cell differences between the models are smaller in the slab ocean simulations, compared to the fixed-SST simulations. The GA7.0 slab ocean circulation cells extend higher and a stronger compared to Flex-UM.

The precipitation and atmospheric energy budget from the slab ocean aquaplanet simulations for Flex-UM and GA7.0 are shown in Fig. 10. Unlike the fixed-SST case, the Flex-UM slab ocean has a single ITCZ and the small double peak seen in GA7.0 fixed-SST simulation is absent in the slab ocean simulations. In the slab ocean Flex-UM case, the LH and LWC is larger compared to the fixed-SST case. The LWC in the Flex-UM slab ocean case has a small dip at the equator, which is not expected, and originates in the LW surface fluxes (not shown). We suspect this LWC dip is the result of the increased equatorial relative



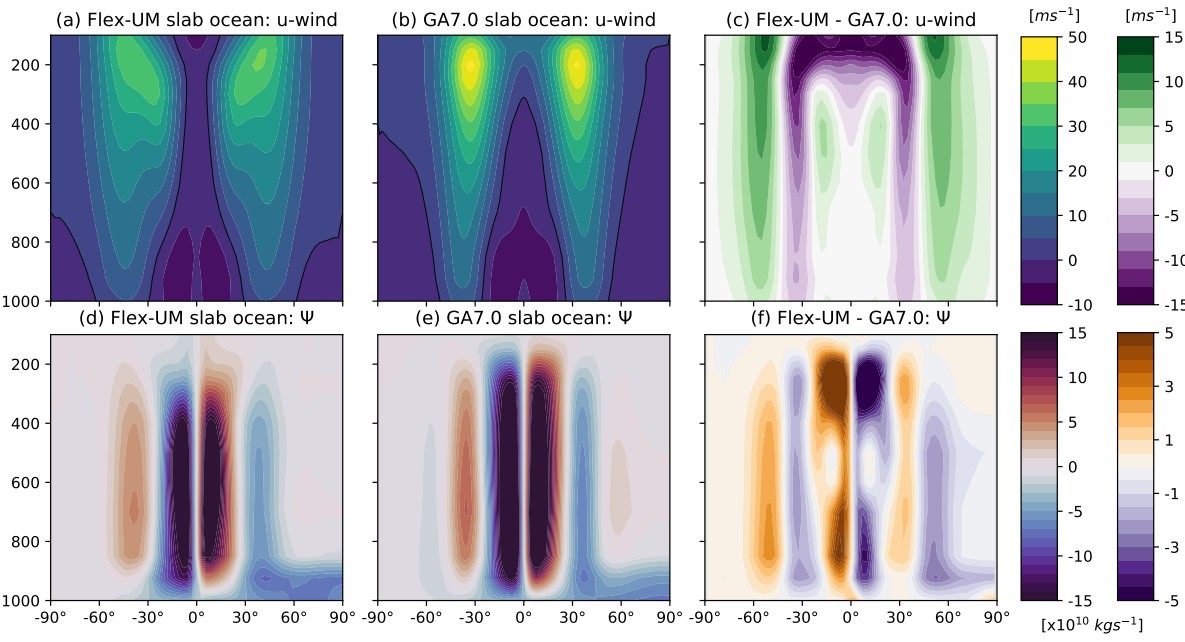

**Figure 9.** As in Fig. 8 but for zonal-mean zonal wind ($u$ in $ms^{-1}$, top panel) and mass stream function ($\Psi$ in $\times 10^{10} kgs^{-1}$, bottom panel).

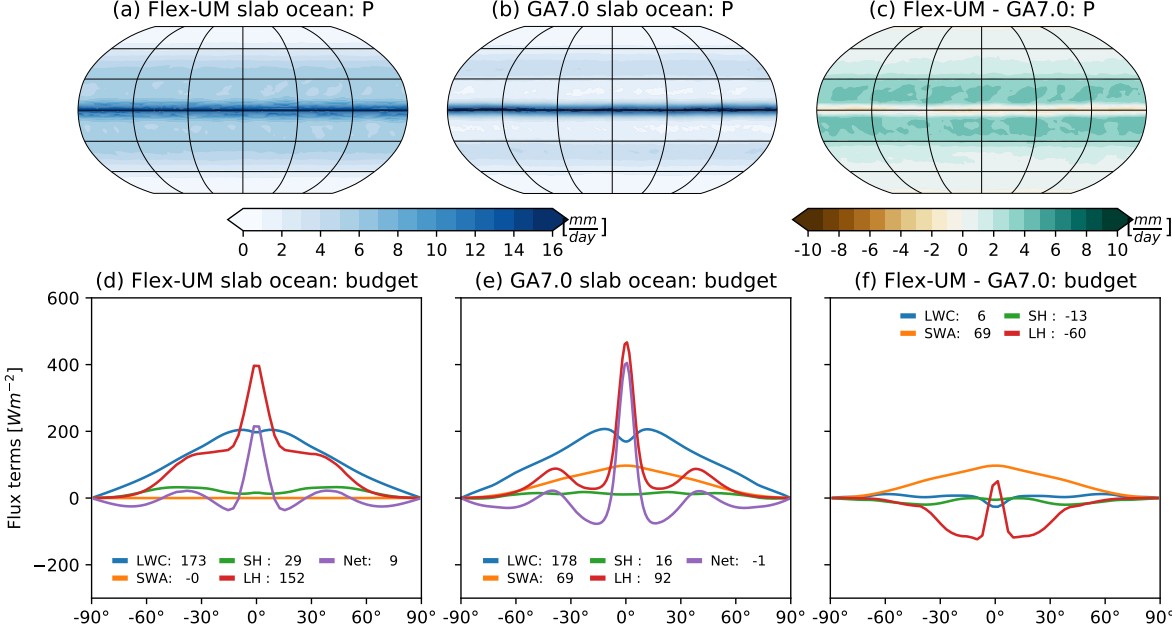

**Figure 10.** As in Fig. 8 but for precipitation ($P$ in $mmday-1$, top panel) and the atmospheric energy budget ($Wm^{-2}$, bottom panel).





humidity in the slab ocean simulation which would increase the optical depth. The SWA in Flex-UM is zero, as discussed in Section 4.1. The Flex-UM slab ocean budget is closer to closing than the fixed-SST case, with a net of 8.8 $Wm^{-2}$. In the slab ocean GA7.0 case, the budget terms are very similar to the fixed-SST case and the budget closed to within -1.5 $Wm^{-2}$. The only shape differences in the budget terms is in the LH where only a single tropical peak is seen.

## 5   Summary and Outlook

A climate model hierarchy is a sequence of models connecting idealised to comprehensive models, via a series of intermediate complexity models. Climate model hierarchies are grounded in truth through comparisons to observations. We use comprehensive climate models to *simulate* the Earth system and idealised models to *understand* fundamental processes within the Earth system. Climate model hierarchies connect our robust physical laws to our complex reality (Maher et al., 2019).

The Met Office UM is a world-leading atmospheric weather and climate model capable of modelling the full complexity of the Earth system and more idealised configurations such as the Newtonian relaxation in the Held-Suarez test case or radiative convective equilibrium. However, the are very few idealised parameteristion options within the UM, limiting the broad use of hierarchical modelling for the UM. In this study we introduce the Flexible modelling framework of the UM – Flex-UM– which greatly enhances the climate model hierarchy capabilities for the UM by include multiple new and adapted idealised parameterisations. These include newly implemented idealised parameterisations of convection, large-scale precipitation, radiation, boundary layer, and SST boundary conditions (using both a fixed-SST and slab ocean configurations). These parameterisations and surface conditions were developed by replicating the GFDL moist physics aquaplanet (Frierson et al., 2006; Frierson, 2007). We have implemented Flex-UM in a modular way, so that one or more schemes can be altered at a time. Hence Flex-UM is a framework rather than a specified model configuration.

We have compared Flex-UM to ERA5, the Isca modelling framework (based off the GFDL aquaplanet) and the comprehensive UM Global Atmosphere 7.0 configuration (GA7.0). Flex-UM captures the general structure of the ERA5 temperature, relative humidity, atmospheric circulation, and precipitation patterns when viewed through the lens of comparing an idealised model to reanalysis. The slab ocean configuration of Flex-UM is similar to the Isca slab-ocean configuration, validating the implementations of the parameterisations within Flex-UM. Compared to Isca, Flex-UM is cooler, has higher relative humidity and a less pronounced subtropical precipitation minimum and storm track maximum. For both the slab ocean and fixed-SST simulations, Flex-UM and GA7.0 are broadly similar, though Flex-UM is cooler, has higher relative humidity, and weaker circulation. The Flex-UM slab ocean configuration generates a single ITCZ, while the fixed-SST case produces a double ITCZ. The double ITCZ is relatively common for aquaplanet configurations (Williamson et al., 2013) and is a well known problem in comprehensive GCM (Tian, 2015; Zhang et al., 2015).

Unlike Isca and GA7.0, the Flex-UM atmospheric energy budget does not close to within a few $Wm^{-2}$ as we had hoped, but rather it gains energy by 9 $Wm^{-2}$ in the slab ocean case and 11 $Wm^{-2}$ for the fixed-SST case. Further work is ongoing to close the energy budget. In the current implementation of Flex-UM, we have not included shortwave absorption within the grey radiation scheme. In Frierson's original development of the GFDL aquaplanet in Frierson et al. (2006) there was





no shortwave absorption and this was later implemented in Frierson (2007). For the next configuration of Flex-UM, we will include shortwave absorption in the downward flux calculation consist with (Frierson et al., 2006). Perhaps this will improve the Flex-UM atmospheric energy budget, which will also be a focus for the next model release. A final feature to add to the next configuration of Flex-UM is to include a cloud scheme.

The goal for developing Flex-UM was to broaden the climate model hierarchy capabilities within the UM. Having achieved this goal, the next natural question is to consider what can Flex-UM be used for? The slab ocean configuration has already been used for simulating terrestrial exoplanets (Boutle et al., 2017) and the simplified Betts-Miller convection scheme has recently been used for a convection scheme intercomparison by Hwong et al. (2021). The parameterisations of Flex-UM will play a role in developing and evaluating the next generation of dynamical core for the UM (named LFRic after Lewis Fry Richardson)

and in evaluating the next generation of convection scheme within the UM (named CoMorph). In addition to supporting model development and evaluation, Flex-UM can also be used to address bold science questions within an easier to understand setting. Answering bold science questions is a fundamental motivation for using climate model hierarchies and Flex-UM enhances this capability within the UM.

*Code and data availability.*   The UM code and it's configuration files are subject to Crown Copyright. A licence for the UM can be requested

from https://www.metoffice.gov.uk/research/approach/collaboration/unified-model/partnership. The source code for Flex-UM will be integrated into the next model release of the UM (version 12.1) and will be available at https://code.metoffice.gov.uk/trac/um/browser (to access this link you first need to apply via the link above). The Flex-UM simulations in this study were completed using modifications to the UM version 11.7 and the GA7.0 simulations were completed using version 11.6. The simplified Betts-Miller scheme and slab-ocean code used in this study are already on the UM code trunk at version 11.7 as part of the Trac code management system (see link above). The grey

radiation scheme, boundary layer code, and large-scale precipitation are currently under review and will be made available as part of the next UM release version 12.1. The configuration files used in this study are available at https://code.metoffice.gov.uk/trac/roses-u/browser for the UM suite IDs: Flex-UM fixed-SST (u-cc036), Flex-UM slab ocean (u-cc037), GA7.0 fixed-SST (u-cc038) and GA7.0 slab ocean (u-cc039). The model output for Flex-UM and GA7.0 can be downlowned at https://doi.org/10.5281/zenodo.5051874. The Flex-UM and GA7.0 data can also be downloaded directly from the Met Office archive using these suite IDs. The Isca model output can be down-

load at https://doi.org/10.5281/zenodo.5017471. The postprocessing and python scripts used in this manuscript can be downloaded from https://doi.org/10.5281/zenodo.5005617.

*Author contributions.*   PE lead the development and evaluation of Flex-UM, contributing the majority of the code for parameterisations and the SST configurations. PM implemented the Simple Betts-Miller convection scheme and assisted in model evaluation. PE performed the Flex-UM and GA7.0 simulations. PM performed the Isca simulations and downloaded the ERA5 data. PM post-processed all the data and

produced the figures in this manuscript. PM wrote this manuscript with technical support from PE.



*Competing interests.* The authors declare that they have no conflict of interest.

*Acknowledgements.* ERA5 data was provided by the Copernicus Climate Change Service (2017) (https://cds.climate.copernicus.eu/cdsapp# !/home, last access: 10 March 2021). The atmospheric energy budget python code is publicly available at https://github.com/penmaher/ radiation_budget. The mass stream function python code is also publicly available at https://github.com/penmaher/streamfunction. The Isca
5  climate model is publicly available at https://github.com/ExeClim/Isca. The simulations in this study were generated using Isca commit ID 66a50d9 (commit date 21 May 2021). PM is funded by the UK Natural Environment Research Council under grant NE/N013123/1 and NE/T003863/1 as part of the ParaCon programme.



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
