# Peer review of "The Flexible Modelling Framework for the Met Office Unified Model (Flex-UM, using UM 12.0 release)"

_Geoscientific Model Development, 2021_

## Referee Comment (RC2)

**Review of the GMDD manuscript**

The Flexible Modelling Framework for the Met Office Unified Model (Flex-UM, part of the UM 12.1 release)

Authors: Penelope Maher and Paul Earnshaw

**Summary:**
The manuscript describes a flexible modelling framework (called Flex-UM) which broadens the climate model hierarchy capabilities of the U.K. Met Office's Unified Model (UM). Simplified physical parameterizations were added to Flex-UM which is part of an upcoming UM release. The parameterizations were originally designed by Frierson et al. (2006) and Frierson (2007) for an idealized moist version of the (now legacy) spectral transform model of the Geophysical Fluid Dynamics Laboratory (GFDL). In particular, simplified schemes for convection, large-scale precipitation, radiation, boundary layer mixing, and sea surface temperature (SST) boundary conditions were included. The purpose of the manuscript is to compare Flex-UM to (a) ERA5 reanalysis data, (b) aquaplanet simulations with another simplified modeling system called Isca, and to (c) comprehensive simulations of the UM (version GA7.0) in an aquaplanet configuration with both a slab-ocean and fixed SST boundary condition. The manuscript thereby aims at documenting the flow characteristics of Flex-UM, judging their realism, and presenting the results as a benchmark calculation that can be used by others for model intercomparisons.

The manuscript is purely descriptive and points out the similarities and differences between the ERA5 data, Flex-UM, Isca and UM GA7.0 in both the slab ocean and prescribed SST mode. No attempt is made to shed further light on the processes that lead to the differences. This approach is acceptable since the focus of the manuscript lies on the documentation of Flex-UM. In general, the manuscript is well written. However, it fails to serve as a benchmark for others due to the poor quality of the figures and some missing pieces of information, such as the exact configuration of the simplified Betts-Miller (SBM) scheme, and the omission of the surface fluxes in the description. There are also some minor inconsistencies, such as unit mismatches in the definition of the latent heat flux. These are outlined below. The major deficiency of the manuscript (but potentially easy to correct) is the poor choice of the color schemes (often white to dark blue or dark blue to dark red which too many shades) which makes it impossible to clearly see the data. This is especially true in a printed copy of the manuscript. In addition, it is clear from many of the figures like the relative humidity and streamfunction figures (including their difference plots) that the chosen min/max ranges for the color schemes are inadequate and that the colors saturate. This fails to show the actual circulation data, which is the main message of the manuscript. All figures need to be replotted. The main goal needs to be the readability of the data, including a display of their actual min/max range. Once corrected, the manuscript will be a valuable addition to the literature that describes model hierarchies.

**Specific comments:**
1) Replot all figures with adequate min/max ranges (capturing the actual data ranges, avoid the large areas with saturated colors) and select clearly distinguishable color schemes.
2) Page 1, line 8, typo: should read 'Geophysical Fluid Dynamics Laboratory'
3) Page 4, line 14: the surface fluxes need to be included in the list, and the treatment of the surface fluxes needs to be described in section 2.

4) Page 4, line 34: Frierson (2007) presents the SBM scheme in many configurations, e.g., with the 'shallower shallow convection scheme', 'no shallow' or 'SBM with qref'. In addition, Frierson explores various parameter ranges for the tuning parameters 'relative humidity threshold' and 'relaxation time scale'. The exact choices for the Flex-UM simulations need to be added to the manuscript.

5) Page 5, line 12: which Frierson model is meant here? None of the listed Frierson papers use a fixed SST. Define the latitude symbol in the definition of the SST. It would be helpful to see the actual SST profile to have a better understanding how it compares to the more standard SST profiles listed in the aquaplanet paper by Neale and Hoskins (2000).

6) Page 5, line 25 and page 6, line 8: Flex-UM uses a 2.5 slab ocean depth, Frierson et al. (2006) selected a 2.4 m slab ocean depth. Why was a different h value chosen in Flex-UM? The manuscript claims that it follows Frierson et al. (2006).

7) Page 6, lines 27 and 28: the authors mean 'extrapolation', not interpolation.

8) Page 7, provide details how the sensible heat (SH) and latent heat (LH) fluxes are defined and computed for Eq. (1) (see also comment 2) which highlights the omission of the surface fluxes in the description).
It is stated that $LH = L_c P$ where P is the precipitation. Correct this to 'precipitation rate'. It is true that ideally the surface evaporation is balanced by the precipitation when averaged over long time periods. Did the authors check whether this is true for Flex-UM? There is a unit mismatch here since P is displayed in units of mm/day (m/s) in all figures. However, the rain rate P in the equation needs units of $kg/(m^2 s)$ which when multiplied with the latent heat of condensation $L_c$ (J/kg) lead to units of $W/m^2$ for the energy budget. Use this opportunity to also clarify the apparent sign inconsistency between the definitions of the evaporative fluxes:
$E = \rho_a C |v_a| (q_a - q^*_a)$    (Frierson et al. (2006), Eq. (11)), used in Flex-UM?
$E = \rho_a C |v_a| (q^*_a - q_a)$    (Vallis et al. (2018), Eq. (10), defined for the Isca model)

9) Caption of Figure 1: The black contour in (c) on top of the dark blue is invisible in a printed copy.

10) Page 10, line 13: when computing the mass stream function was the time-mean zonal-mean of v and $\omega$ computed first before the integration of the velocity components?

11) Page 10, figure 2 and figure 3: it seems wrong that the Isca model shows a hemispheric asymmetry in the gray (no-data) area near the surface in the Southern Hemisphere. What is the reason for this or is this a plotting error? A 10-year average should not have such a clear asymmetry. This question also addresses the hemispheric asymmetry in the stream function plots in Figs. 3, 6 and 9.

12) Precipitation rate plots (Fig. 4, 7, 10): since there are almost no longitudinal variations in P in these plots, it would be a lot clearer to show the zonal-mean P as line plots instead. This will also make it easier to see the min/max ranges of P and the P differences.

13) Captions of Figs. 4, 7, 10: typo, should read mm day$^{-1}$

---

## Author Comment (AC1)

**Response to reviews for GMD manuscript gmd-2021-193 Maher and Earnshaw 2021**

We kindly thank the reviewers for the time and effort they invested in reviewing our paper. In this response to reviewers, comments from the reviewer will be in italics and our response without italics. References to line numbers (i.e page 2 line 20 will be written P2 L20) in the reviewers comments will correspond to the original submission and all other line numbers will correspond to the revised manuscript. All Zenodo archives have been updated to reflect changes made during review.

**Response to Reviewer 1**

Thank you for the positive description of our motivation and summary of the manuscript. We are pleased to see that with some minor adjustments to the manuscript that you think the paper is suitable for publication. The main critique raised is the lack of discussion in the manuscript. We have remedied this and will describe it in more detail below.

**Response to specific comments from Reviewer 1**

1. *Introduction: As I understand Flex-UM was designed to replicate the GFDL idealised model, which seems to be a benchmark for this model type. It is not quite clear to me what this exactly means? Flex-UM has the same dynamical core as GA7.0, namely ENDGame, but simplified physical parameterisations. In which respect is it directly comparable to the GFDL aquaplanet model? And how is this comparability achieved?*

   Good point. You are correct that the dynamical core of Flex-UM and GFDL are different. What we mean here is that the parameterisations, boundary conditions and horizontal/vertical resolutions have been made as similar as possible. To make this clearer P2 L20 now reads: "Flex-UM was designed to replicate the atmospheric physics of the Geophysical Fluid Dynamic Laboratory (GFDL) simplified moist physics aquaplanet (Frierson et al., 2006, 2007), which is a benchmark intermediate complexity climate model."

2. *Model name and acronyms: I find the amount of used acronyms and model names for someone who is not familiar with the UM-family quite confusing (UM, UKESM1, GC3.0, GA7.0, HadGEM-GC3.0, . . . ). I know traceability of model versions/configurations is important to GMD, but I think it does not necessarily increase readability. Maybe the authors have a sketch at hand which gives an overview of the UM family and could be shown in the paper (as Appendix)?*

   We agree that too many acronyms do impact readability. We feel the use of "UM" for the unified model and GA7.0 for the model version are necessary for clarity in the paper. But we agree about UKESM1 and GC3.0 should have limited use. We have removed the unnecessary use of UKESM1 and GC3.0 where ever possible. We considered removing the acronym HadGEM3-GC3.0, but as this is the CMIP named version of the model we felt it was suitable to include in the paper. Only the UM and GA7.0 are routinely used in the paper and all other UM-family acronyms are limited to the last 2 paragraphs of page 2. We are not aware of a UM model schematic that would be suitable to include in the paper. We agree such a schematic would be helpful but generating it would require consultation and approval from the Met Office (as branding is important). As such, we have decided not to include a model schematic.

3. *Sect. 3: I would suggest to add a table summarizing the performed model experiments and the key parameters of each model configuration, e.g. resolutions.*

   Table 1 has been added to the manuscript which includes the vertical and horizontal resolutions and the length of datasets.

4. *Different resolutions: On page7, line 8-9 the authors state that "key differences in the climatologies are highly unlikely to be the result of resolution differences". Would it be possible to run Flex-UM and GA7.0 at the same resolution to underpin this statement?*

   While we agree that the ideal scenario would be that the Flex-UM and GA7.0 should be run at the same resolution, some components of the UM system make this very difficult. GA7.0 has been very heavily tuned to run with a specific scientific configuration that requires it to be run with a high model top ($\sim 80\text{-}85$km). This includes meso- and strato-spheric parameterizations, and input forcing datasets. Flex-UM has initially been set up without these components of the configuration which limits the practical model top of the system ($\sim 40$km) as there are no constraints on the behaviour of the upper stratosphere and above (e.g. non-orographic gravity waves). The decision was made that the GA7.0 aquaplanet should be run at the same resolution as the full production climate model to provide the "gold standard" against which the Flex-UM and Isca models will be measured. This accepts that the latter two models will be limited in their resolution by the physics that will be implemented, but making sure that the resolutions they do have will be very similar.

5. *P10, L15 onwards: I find this explanation too simplified and sketchy. How does the limited resolution impact the distribution of v and omega, and what integration artefacts are meant?*

   Please see discussion on this issue at the end of the review.

6. *Fig. 4a: Flex-UM seems to show a kind of "wavy" structure in the precipitation climatology in the subtropics, which is not apparent in Isca (or not visible in Fig. 4b). Any explanation for that behaviour?*

   The precipitation in Flex-UM between $30°$ in both hemispheres does show some structure with pocks of weak precipitation in the climatology (giving a wavy appearance in the difference plot). It is not clear to us why there are preferred locations for reduced precipitation in these regions.

7. *P11/12, discussion of Fig. 4f: The authors rate the missing solar absorption in Flex-UM as key difference to Isca. Are the shown differences in the climatologies consistent with the SWA of Flex-UM being zero? How does the missing SWA impact the model performace in general? I would like see some discussion of the (potential) underlying reasons for the presented model differences, even if the authors can only speculate. This holds also for other sections. In the conclusions the authors state that including solar absorption might improve the atmospheric energy budget. It would be interesting to see which changes the authors expect from this modification. And if it will not help to close the energy budget, what would be the next steps.*

   We have added a new paragraph on this. Please see the first paragraph on page 14.

8. *Regarding the comparison with ERA5: I think it is clear that zonally symmetric aquaplant models do not give the same results as ERA5, so what is the idea of comparing the idealized models to ERA5?*

   We absolutely agree that the idealised models like these are not going to look the same as reanalysis. The motivation for comparing to ERA5 is more to ground the reader in what is truth. By showing what the real world looks like, we can then see if the idealised models are doing a good job or not. To make this clearer we added the following to page 8: "We do not expect that Flex-UM will replicate all of the structure seen in ERA5, however, ERA5 climatologies provide a reference point so that we can assess the realism of the idealised models in this paper."

9. *Sect. 4.3: For the discussion of the differences between the fixedSST and slab ocean experiments it might be useful to show the difference in the surface temperatures. Can different surface temperature patterns be used to explain differences between the fixedSST and slab ocean experiments?*

   Figure 1 has been added to the manuscript, showing the slab and fixed SSTs for each experiment and their differences. Please see the new paragraph at the end of page 6. The SSTs are the only difference between the simulations for the fixed SST and the slab ocean experiments for each model. The SSTs would drive first order changes in the model climatology, for example warmer equator would generate more precipitation. However, the SSTs would also drive second order

changes too, such as changes to the equator to pole temperature gradients would change the global circulation that in turn would impact precipitation. As such, the SSTs alone can not be attributed to differences in model climatologies but they certainly do play a key role.

**Response to Technical comments**

1. *P6, L9: There are some closing parentheses ")" missing.*
   Changed as suggested.

2. *P6, L28: "Interpolation is prone to[?] errors..."*
   Added 'to'.

3. *Fig. 1: Why are there no grey contours in the NH? Rocky Mountains, Tibetean Plateau,...? Is the masking done before or after zonal averaging?*
   The data over the Rocky Mountains, Tibetean Plateau etc are masked. The zonal average is taken after the data is masked and hence the data at these locations is not included in the zonal mean.

4. *Fig. 2 and 3: It looks like the grey masking is not the same in the middle and right plots. Shouldn't it be identical?*
   The left and middle plots are on the native model grids. The resolution of the models is the same but the grids are not. So that the models could be plotted as differences, they were regridded to a common grid. As such, the right plot will not have the same masking as the left and middle plot.

5. *P11, L8: closing ")" missing: ... and in the LH of Fig. 4(e)).*
   Changed as suggested.

6. *Fig. 4f: I think this plot shows Isca – Flex-UM, and not Flex-UM – Isca as written in the header. The SWA difference is +59 W/m2, but since the SWA of Flex-UM is zero it should be negative for the difference Flex-UM – Isca.*
   Very well spotted. You are correct. This has been fixed and the title and data are now consistent.

7. *P13, L4: I think it must read "... at a lower altitude compared to GA7.0..."*
   Well spotted. Changed as suggested.

8. *Fig. 7f: Same as Fig. 4f, GA7.0 – Flex-UM*
   This has been fixed.

9. *Fig. 10f: Same as Fig. 7f, GA7.0 – Flex-UM*
   This has been fixed.

10. *P15, L22: "It has also been shown that[?] the single ..."*
    Changed as suggested.

11. *P15, L24: "... long standing problems[?]..."*
    Changed as suggested.

12. *Caption Fig. 4, 7, 10: mmday-1, -1 should be superscript*

   Changed as suggested.

13. *P18, L12: "However, there[?] are very few. . . "*

   Changed as suggested.

14. *P18L14: ". . . by including[?] multiple new. . . "*

   Changed as suggested.

15. *P19L2: citation: (Frierson et al., 2006) -> Frierson et al. (2006)*

   Changed as suggested.

16. *P19L14: "it's configuration", it's -> its*

   Changed as suggested.

**Response to Reviewer 2**

The main criticism of the manuscript was the style of figures presented in the manuscript. Key issues identified were: colour schemes, the min/mas range not suitable (as it saturates) in the relative humidity and stream function plots.

**Response to specific comments from Reviewer 2**

1. *Replot all figures with adequate min/max ranges (capturing the actual data ranges, avoid the large areas with saturated colors) and select clearly distinguishable color schemes.*

   We do not agree that all of the min/max ranges fail to capture the actual data. Our motivation for limiting RH at 100% was to show where the data is super saturated. We acknowledge this means anything above 100% can't be distinguished. This was a style choice but we can appreciate this will not suit all readers. To remedy this we have extended the RH key to 140% for all plots (including for ERA5 for consistency - but we acknowledge this does look strange out of context). We also expanded the key for the stream function plots. We have retained the min/max ranges for T, u and P as these capture the range of the data.

   In terms of colour schemes, the colour palettes we have selected are standard for these types of climatologies so we have retained our original colour schemes.

2. *Page 1, line 8, typo: should read 'Geophysical Fluid Dynamics Laboratory'*

   Changed as suggested.

3. *Page 4, line 14: the surface fluxes need to be included in the list, and the treatment of the surface fluxes needs to be described in section 2.*

   The description of the surface fluxes has been added to the paper, along with a reference to the JULES paper where the surface exchange scheme is officially documented. See paragraph 3 on page 5.

4. *Page 4, line 34: Frierson (2007) presents the SBM scheme in many configurations, e.g., with the 'shallower shallow convection scheme', 'no shallow' or 'SBM with qref'. In addition, Frierson explores various parameter ranges for*

*the tuning parameters 'relative humidity threshold' and 'relaxation time scale'. The exact choices for the Flex-UM simulations need to be added to the manuscript.*

Excellent points. This is now included in the manuscript as follows: "The "shallower" shallow convection of Frierson (2007) is used, consistent with Isca. The two tunable parameters for the convection scheme are the critical relative humidity and the relaxation time, which are set to 70%, and 2 hours, respectively, consistent with the default parameters of Isca."

5. *Page 5, line 12: which Frierson model is meant here? None of the listed Frierson papers use a fixed SST. Define the latitude symbol in the definition of the SST. It would be helpful to see the actual SST profile to have a better understanding how it compares to the more standard SST profiles listed in the aquaplanet paper by Neale and Hoskins (2000).*

For clarity, the manuscript now reads: "This SST profile and its default values are the same as the initial SST profiles in the Frierson model and Isca."

The latitude symbol is now defined.

While we appreciate that the collection of more standard SST profiles from Neale and Hoskins (2000) are more widely used for fixed SST simulations, we chose to use the SST profile that was the initial value in the original GFDL model. To introduce the Neale and Hoskins (2000) SST profiles in this paper we believe would create an unnecessary confusion as to why we did not use them. It is entirely possible that future work on this model could use the more standard profiles if it is deemed they are more appropriate.

6. *Page 5, line 25 and page 6, line 8: Flex-UM uses a 2.5 slab ocean depth, Frierson et al. (2006) selected a 2.4 m slab ocean depth. Why was a different h value chosen in Flex- UM? The manuscript claims that it follows Frierson et al. (2006).*

The slab ocean in Flex-UM is defined by it's heat capacity (10**7 J K-1 m-2) and this is in line with the configuration allowed for the surface exchange scheme in JULES. This appears to be how the slab ocean in Frierson et. al. (2006) is defined in equation (1) and from the parameter in Table 1. The value of 2.5m was taken as an approximation of the equivalence of this heat capacity to ocean depth and was in line with the value used in the Isca model.

7. *Page 6, lines 27 and 28: the authors mean 'extrapolation', not interpolation.*

Changed as suggested

8. *Page 7, provide details how the sensible heat (SH) and latent heat (LH) fluxes are defined and computed for Eq. (1) (see also comment 2) which highlights the omission of the surface fluxes in the description). It is stated that $LH = L_c P$ where P is the precipitation. Correct this to 'precipitation rate'. It is true that ideally the surface evaporation is balanced by the precipitation when averaged over long time periods. Did the authors check whether this is true for Flex-UM? There is a unit mismatch here since P is displayed in units of mm/day (m/s) in all figures. However, the rain rate P in the equation needs units of $kg/(m^2 s)$ which when multiplied with the latent heat of condensation $L_c$ (J/kg) lead to units of W/m2 for the energy budget. Use this opportunity to also clarify the apparent sign inconsistency between the definitions of the evaporative fluxes:*
*$E = \rho_a C |v_a|(q_a - q^* a)$ (Frierson et al. (2006), Eq. (11)), used in Flex-UM?*
*$E = \rho_a C |v_a|(q^* a - q_a)$ (Vallis et al. (2018), Eq. (10), defined for the Isca model)*

The first use of precipitation in this context was changed to precipitation rate for clarity and then the remainder of the manuscript uses the term precipitation under the assumption it is the precipitation rate.

The Flex-UM convention for surface fluxes is upwards, as seen in the formulations now included in the paper on page 5, so they are consistent with the Isca model and not with equation (11) published in Frierson et al. (2006).

In terms of the P-E values, these have now been added to the manuscript. Please see the last paragraph of section 3.2.

When computing the energy budget, precipitation was used in its flux form (W/m2) while the plotted climatologies for precipitation use the precipitation rate (mm/day).

9. *Caption of Figure 1: The black contour in (c) on top of the dark blue is invisible in a printed copy.*

   The contouring link thickness was increased and is now grey.

10. *Page 10, line 13: when computing the mass stream function was the time-mean zonal- mean of v and w computed first before the integration of the velocity components?*

    The stream function is computed on zonal mean data (not the time-mean zonal-mean) and then the time mean of $\psi$ is computed. We have changed the text to read: "The two integrations are used to compute $\psi$ for each month before computing the time mean climatologies."

11. *Page 10, figure 2 and figure 3: it seems wrong that the Isca model shows a hemispheric asymmetry in the gray (no-data) area near the surface in the Southern Hemisphere. What is the reason for this or is this a plotting error? A 10-year average should not have such a clear asymmetry. This question also addresses the hemispheric asymmetry in the stream function plots in Figs. 3, 6 and 9.*

    While the theoretical model configuration is equatorially symmetric, the numerical implementation can give rise to a level of asymmetry. This will be seen in any image of an instantaneous field of the simulation. We would hope that for long time scale means these asymmetries would become part of the numerical noise and not apparent on any image. However the processing of these fields involves taking account of when the pressure level plotted is below the surface, and then applying a threshold for how long a given point is below the surface. If the value of this time fraction is very close to the threshold then it is very possible to get asymmetric behaviour in the plots. We understand it is not desirable, but it is how the data presents itself.

12. *Precipitation rate plots (Fig. 4, 7, 10): since there are almost no longitudinal variations in P in these plots, it would be a lot clearer to show the zonal-mean P as line plots instead. This will also make it easier to see the min/max ranges of P and the P differences.*

    We agree there is minimal longitudional variation in the P plots. The motivation for showing P as maps is that we feel the difference plots are more meaningful as a map than as a zonal mean, and the climatological maps give context to the climatology differences. The zonal mean P is plotted in the budget plots on the same figures.

13. *Captions of Figs. 4, 7, 10: typo, should read mm day-1*

    Changes as suggested

**What is the artifact in $\psi$ referred to in the manuscript?**

Please excuse the low quality figures and screen shots. These images were made for testing the code, hence they are not paper quality images. The code to compute the stream function has been made publicly available and can be found here: https://github.com/penmaher/streamfunction.

$\psi$ is computed by integrating omega and the v component of wind. The climatologies of these fields are shown in Fig.1.

The integration was computed using left and right Riemann sums. Python can compute these using the cumtrap function (part of the scipy library scipy.integrate.cumtrapz) , however, this method does not handle masked data and so it was not used (it was used to check the Riemann sums only). The left and right Riemann sum for the v wind integrated with respect to pressure are shown in Fig. 2. The average of the two Riemann sums (bottom right of Fig. 2) does not have an integration artifact over the NH polar region. The left and right Riemann sum for omega integrated with respect to latitude are shown in Fig. 3. In the left, right, and average Riemann sum plots for omega the NH polar region has an unrealistic circulation (what we

[Figure]

**Figure 1.** v component of the wind (left) and omega (right) for the fixed SST Flex-UM experiment.

[Figure]

**Figure 2.** Riemann sums for the v wind integrated wrt pressure. Top panel (L and R): left Riemann sum and right Riemann sum. Bottom pane; (L and R) integration using the cumtrap python method and the average of the left and right Riemann sums.

refer to in the manuscript as an integration artifact). The stream functions is computed using the averaged Riemann sum, see Fig. 4 (left panels), also has the artifact and hence it is also in their mean (Fig. 4 right panel). We could have selected to only plot $\psi$ computed using the integration of v component, however we felt this was not a correct representation of $\psi$.

[Figure]

**Figure 3.** As in Fig. 2 but for the omega integrated wrt latitude.

[Figure]

**Figure 4.** Top left: $\psi$ computed using the averaged Riemann sums integrated wrt omega. Bottom left: $\psi$ computed using the averaged Riemann sums integrated wrt v-wind. Middle right: the average of the two $\psi$ in the left plots (this figure is an early draft of the fixed SST Flex-UM data which is Fig. 7(d) in the manuscript.